High variability of dung beetle diversity patterns at four mountains of the Trans-Mexican Volcanic Belt

Arriaga-Jiménez Alfonsina 1 2
Rös Matthias 3 iguarana@gmail.com
Halffter Gonzalo 4
1 CIIDIR Oaxaca, Instituto Politécnico Nacional , Oaxaca , Mexico
2 Laboratoire de Zoogéographie, UMR 5175 CEFE, Université Paul Valéry (Montpellier III) , Montpellier , France
3 CONACYT, CIIDIR Oaxaca, Instituto Politécnico Nacional , Oaxaca , Mexico
4 Red de Etoecologia, Instituto de Ecologia, A.C. , Xalapa, Veracruz , Mexico
Andrew Nigel
Electronic publication date: 2018 Feb 27
Publication date: 2018
Volume: 6
Electronic Location ID: e4468
Received 2017 Aug 16; Accepted 2018 Feb 16
Copyright: © 2018 Arriaga-Jiménez et al.
Copyright year: 2018
Copyright holder: Arriaga-Jimenez et al.
License: This is an open access article distributed under the terms of the Creative Commons Attribution License, which permits unrestricted use, distribution, reproduction and adaptation in any medium and for any purpose provided that it is properly attributed. For attribution, the original author(s), title, publication source (PeerJ) and either DOI or URL of the article must be cited.
License URL: https://creativecommons.org/licenses/by/4.0/

Keywords: Mexican transition zone, Compositional similarity, Species distribution patterns, q-Diversity, Horizontal colonization, Environmental instability, Archipelago reserves

Funding: Alfonsina Arriaga from the Consejo Nacional de Ciencia y Tecnología, Mexico 310188 The Université Paul-Valéry Montpellier III ED60-2012 Société Entomologique de France 2012 This work was supported by a doctoral grant to Alfonsina Arriaga-Jiménez from the Consejo Nacional de Ciencia y Tecnología, Mexico (No. 310188). The Université Paul-Valéry Montpellier III provided a doctoral mobility scholarship (ED60-2012), as well as the Société Entomologique de France (Germaine Cousin Grant, 2012). The funders had no role in study design, data collection and analysis, decision to publish, or preparation of the manuscript.

==============================
Insect diversity patterns of high mountain ecosystems remain poorly studied in the tropics. Sampling dung beetles of the subfamilies Aphodiinae, Scarabaeinae, and Geotrupinae was carried out at four volcanoes in the Trans-Mexican Volcanic Belt (TMVB) in the Mexican transition zone at 2,700 and 3,400 MASL, and on the windward and leeward sides. Sampling units represented a forest–shrubland–pasture (FSP) mosaic typical of this mountain region. A total of 3,430 individuals of 29 dung beetle species were collected. Diversity, abundance and compositional similarity (CS) displayed a high variability at all scales; elevation, cardinal direction, or FSP mosaics did not show any patterns of higher or lower values of those measures. The four mountains were different regarding dispersion patterns and taxonomic groups, both for species and individuals. Onthophagus chevrolati dominated all four mountains with an overall relative abundance of 63%. CS was not related to distance among mountains, but when O. chevrolati was excluded from the analysis, CS values based on species abundance decreased with increasing distance. Speciation, dispersion, and environmental instability are suggested as the main drivers of high mountain diversity patterns, acting together at different spatial and temporal scales. Three species new to science were collected (>10% of all species sampled). These discoveries may indicate that speciation rate is high among these volcanoes—a hypothesis that is also supported by the elevated number of collected species with a restricted montane distribution. Dispersion is an important factor in driving species composition, although naturally limited between high mountains; horizontal colonization events at different time scales may best explain the observed species composition in the TMVB, complemented by vertical colonization events to a lesser extent. Environmental instability may be the main factor causing the high variability of diversity and abundance patterns found during sampling. Together, we interpret these results as indicating that species richness and composition in the high mountains of the TMVB may be driven by biogeographical history while variability in diversity is determined by ecological factors. We argue that current conservation strategies do not focus sufficiently on protecting high mountain fauna, and that there is a need for developing and applying new conservation concepts that take into account the high spatial and temporal variability of this system.

Introduction

The importance of studying high mountain ecosystems in order to understand biodiversity patterns and evolutionary processes is widely recognized (Körner, 2000; Schmitt, 2009). Mountain tops have different ecological conditions from their surrounding lowlands, and interconnection of their biota is less than in other ecosystems. Vertical colonization (highland assemblages composed by species phylogenetically related with those inhabiting lowlands) and horizontal colonization (colonization of highland assemblages by lineages with a different evolutionary history and origin than those occupying lowlands), as well as speciation, are described as drivers of mountain diversity (Lobo & Halffter, 2000; Escobar, Lobo & Halffter, 2006; Schmitt, 2009; Halffter & Morrone, 2017). Mountain tops can act as refuges for flora and fauna that had expanded during glacial or cooler conditions; as a consequence, their biota exhibits greater differences when compared to that in lower elevations, especially in tropical areas where ecological conditions change dramatically among different altitudes (Körner, 2007).

Research exclusively focused on diversity patterns on tropical mountains at elevations higher than 2,500 MASL is scarce (Mastretta-Yanes et al., 2015). The main focus of diversity patterns in mountains has been directed to altitudinal gradients, where literature is abundant (Hanski & Niemelä, 1990; Kessler, 2000; McCain & Grytnes, 2010; Nogués-Bravo et al., 2008; Rahbek, 1997). Mountain gradients often show higher regional biodiversity due to a high species turnover than areas of the same size in tropical lowlands (Rahbek, 1997). Also, mountains are speciation hotspots with a high degree of endemic species (Halffter, 1987; Marshall & Liebherr, 2000). In recent times, with the Earth facing climate change, species of high mountains are thought to be exposed to greater extinction risk due to a changing ecosystem and upward colonization of species adapted to warmer climates (Cahill et al., 2012).

Altitudinal gradients in the tropics present mostly two patterns: (1) Decreasing species richness with increasing altitude (Alvarado, Escobar & Montero-Muñoz, 2014) or (2) species richness peaks at middle elevations, due to climatic conditions following humidity gradients (Nunes et al., 2016) or due to geometrical constraints, called mid-domain effect (Colwell & Hurtt, 1994; Colwell & Lees, 2000; Colwell, Rahbek & Gotelli, 2004, 2005). These patterns have in common that diversity is lowest at high elevations.

The Mexican transition zone (MTZ) is characterized as an area where biotas with Nearctic and Neotropical (NT) origins overlap. It extends from the south of the United States to the plains of southern Nicaragua (Halffter & Morrone, 2017). Since the land connection made by the Panama Bridge, 3.5 million years ago, mountains in the MTZ were primarily dispersion tracks for the northern fauna adapted to cold conditions (which benefited additionally from quaternary glaciations), whereas the lowlands allowed NT species to pass through northern areas (Halffter, 1987; Halffter & Morrone, 2017). As discussed by Halffter (1976), the high level of endemic entomofauna in the MTZ is considered in part a product of speciation following vertical and horizontal colonization, which maintain well-defined affinities with faunas to the south and north (Halffter, 1964, 1976; Lobo & Halffter, 2000).

The Trans-Mexican Volcanic Belt (TMVB) is an irregular province oblique to the American Average Trench. It crosses Mexico between 19° and 21° northern latitude with a mean elevation of 2,300 MASL at its high plateau, where the eight highest volcanoes of Mexico are distributed, which have peak elevations ranging between 4,100 and 5,600 MASL (Mooser, 1972; Demant, 1978; Ferrari et al., 1999). The TMVB is one of the largest physiographic provinces in Mexico, formed during the Cenozoic (65 Ma to the present), and one of the best studied mountain system of the MTZ (Arroyo-Cabrales et al., 2008). In general, locally low diversity of TMVB has been described, but with a high species turnover and a high degree of endemic species (Munguía, 2004). Nevertheless, not much is known about diversity patterns above 2,500 MASL (Mastretta-Yanes et al., 2015).

Dung beetles are one of the best studied insect groups in Mexico and in the tropics in general; compared to many other insect groups their ecology, natural history, biogeography, and diversity patterns are well known (Halffter, 1991; Halffter & Edmonds, 1982; Halffter & Matthews, 1966; Hanski & Cambefort, 1991; Scholtz, Davis & Kryger, 2009). For these reasons, dung beetles have been used widely as a biodiversity indicator group (Halffter & Favila, 1993; Nichols & Gardner, 2010). Commonly, the term dung beetle is used for the three subfamilies Scarabaeinae, Aphodiinae, and Geotrupinae, all belonging to the Scarabaeoidea superfamily. Whereas Scarabaeinae are most diverse and abundant in the tropics, the two latter groups are more diverse in northern temperate regions (Hanski & Cambefort, 1991). In the MTZ, all groups coincide at the same latitudes, but at different richness and abundance levels depending mostly on the elevation (Halffter, 1987; Halffter & Morrone, 2017). Halffter (1976) proposed and discussed that the entomofauna in the MTZ could not only be separated into Northern or Southern origin, but regarding their dispersal patterns (DP), which were defined as the actual distribution of a group of biota originating in a defined area and coexisting for a long period, thus sharing a common biogeographic history (Halffter & Morrone, 2017). Besides NT and Nearctic DP, the Paleoamerican DP is abundant, which corresponds to species distributed in the zone long before both Americas were interconnected (Halffter, 1987; Halffter & Morrone, 2017). Geotrupinae present Nearctic DP, Aphodiinae Nearctic as well as NT DP, while Scarabaeinae show the highest variety, including NT DP, and different subgroups of the Paleoamerican DP (Halffter & Morrone, 2017). In tropical mountain gradients, Scarabaeinae were featured by patterns already described, with peaks at lowlands or mid-elevations; in contrast, Aphodiinae and Geotrupinae were absent or rare at lowlands or mid-elevations, increasing their diversity at elevations higher than 2,000 MASL or sometimes are exclusively limited to these altitudes (Hanski & Cambefort, 1991).

This work represents the first study of diversity patterns of dung beetles carried out at high mountains, at altitudes between 2,700 and 3,400 MASL. No comparable studies were found in the literature, as high mountains were mostly approached as part of altitudinal gradient studies, where sampling effort at high elevations were comparably low (Davis, Scholtz & Chown, 1999; Lobo & Halffter, 2000; Escobar, Halffter & Arellano, 2007; Herzog et al., 2013).

The aims of this research are: first, to study dung beetle diversity patterns at MTZ high mountains extensively in order to collect representative data on species richness and composition, as a lack of sampling intensity in former studies was identified; second, to analyze compositional similarity (CS) at different spatial scales within and between MTZ mountains; and finally, to provide information that could indicate the probable origin of diversity patterns, focusing on biogeographical factors (e.g., vertical and horizontal colonization, speciation), as well as ecological variables (e.g., elevation, exposure, forest mosaics). We predict that (a) dung beetle alpha and gamma diversity of tropical high mountains in the MTZ will be higher than previously documented in altitudinal gradient studies; (b) CS within mountains will be relatively high, with the expected small differences between sampling sites explained by altitude, cardinal direction, and forest mosaics; and (c) CS between mountains will be low, because each mountain is largely defined by its own history and dung beetle fauna.

Methods

Sampling sites

Four mountains were sampled in the TMVB: La Malinche (4,460 MASL), Cofre de Perote (4,200 MASL), Pico de Orizaba (5,610 MASL), and Sierra Negra (4,580 MASL; mountains are abbreviated as: MA, CP, PO, and SN, respectively; see Fig. 1). The three latter volcanoes separate the Mexican High Plateau from the coastal plains of the Gulf of Mexico (Concha-Dimas et al., 2005). The eastern part of the TMVB is the most recent one, and the peaks studied are modern in their current form. The oldest is the Malinche volcano, followed by Cofre de Perote, Sierra Negra, and Pico de Orizaba (Carrasco-Núñez, 2000; Siebert & Carrasco-Núñez, 2002; Neyra Jauregui, 2012).

Figure 1 Map of the study area.

Map of the study sites in the Trans-Mexican Volcanic Belt (indicated by the rectangle), with the four sampled volcanos (indicated by triangles). MA, La Malinche; CP, Cofre de Perote; PO, Pico de Orizaba; SN, Sierra Negra. The map is based on the digital elevation model for Mexico, provided by INEGI (downloadable at http://www.inegi.org.mx).

Volcanoes’ vegetation between 2,500 and 3,500 MASL is characterized by a high degree of heterogeneity, naturally consisting of a mosaic of pine forest, shrubland, and pastures. The treeline is up to 4,020 MASL (Körner & Paulsen, 2004). Pine forest is dominated by species of Pinus and Abies (e.g., Pinus hartwegii LINDL and Abies religiosa (KUNTH) SCHLTDL & CHAM). They are abundant at all four mountains (Neyra Jauregui, 2012). This forest is characterized by a semi-open canopy which allows sunlight to reach the ground, where shrubs and grasses can grow. Vegetation structure is also heterogeneous in terms of tree age, height, and basal area; and patches where trees at an early stage of succession are frequent.

Due to human influence in the TMVB during the past centuries, high mountain landscapes are modified, and natural vegetation was replaced by crops and pastures for livestock. Milpa (the traditional Mesoamerican non-intensive polyculture system of maize, beans, and other plants) is the principal agroecosystem at the region, at higher altitudes non-intensive potato crops, and at lower sites more intensive corn crops are recurrent. Locally, natural pastures are used for cattle and goat grazing. Forest extension differs among volcanoes; MA has the greatest forest area, followed by CP and PO, while SN presents the least extent forest.

Sampling design

At each mountain, four sampling sites were chosen at two different altitudes (∼2,700 and ∼3,400 MASL) and on two directions (windward/east and leeward/west). Due to the heterogeneity of the vegetation, no homogeneous sampling sites could be established. Instead, sampling sites reflected vegetation variability of the forest–shrubland–pasture (FSP) mosaic.

At MA the two upper sampling sites were dominated by forest, whereas the FSP mosaic prevailed in lower sites. At higher sites at CP, a forest–shrubland (FS) mosaic was abundant, where the forest dominated larger parts of it, with pasture present only at the west. The eastern lower sampling site was formed by a well-conserved forest, whereas the western site had a FS mosaic (without dominance of any), with pastures nearby. The FSP mosaics at upper sites in PO were shrubland–pasture dominated in the west, and more forest dominated in the east. The western lower site consisted of an open forest with pastures in the treeless areas. At the eastern lower site the FSP mosaic was dominated by forest, surrounded by milpas. At SN all sites were formed by FSP mosaics.

Sampling was accomplished during the rainy seasons (June–August) from 2011 to 2013. Ten pitfall traps baited with human dung were placed at each site, separated at least by 50 m (Larsen & Forsyth, 2005), left for 48 h and repeated once. Traps were placed exclusively in the natural vegetation types (forest, shrubland, or pasture), but not in crops or pastures used for cattle grazing. The sampling effort was the same, totaling 240 trap days per mountain. This protocol was followed each year, intercalating months for each mountain (e.g., MA was sampled in June 2011, July 2012 and August 2013, see Table S1). Later, dung beetle composition and abundance of each site were recorded. Direct sampling was done during the three years, collecting inside gopher nests and in livestock excrements, in order to capture species not attracted by baits. Nevertheless, these species and individuals were not used in the analysis, but results are included in the discussion. Field experiments were performed with a permission of the Secretaria del Medio Ambiente y Recursos Naturales, Mexico (FAUD-0018). Species were identified by experts for each of the subfamilies: Marco and Giovanni Dellacasa (Universita di Pisa) determined Aphodiinae, Mario Zunino (Universita di Urbino) and Gonzalo Halffter the Scarabaeinae and Geotrupinae species. All the vouchers were deposited in entomological collections: Dellacasa (Genova, Italy), Morón (Xalapa, Mexico) and Halffter (Xalapa, Mexico) personal collections, as well as a reference collection in the Colección Entomológica IEXA, INECOL (Xalapa, Mexico).

Analysis

Unweighted diversity partition based on Hill Numbers proposed by Jost (Hill, 1973; Jost, 2006, 2007) was applied using orders q = 0 and 2, where the first is equal to species richness, the latter manifests patterns for abundant species, and their unit is the effective number of species. The formula is: qD=(∑i=1Spiq)1/(1−q)

These methods are now widely used and have been described in detail many times (Arroyo-Rodríguez et al., 2013; Jost, 2007; Martínez et al., 2009; Murillo-Pacheco et al., 2016). In the unweighted form, mean relative abundance is used to determine gamma diversity. Beta diversity is the quotient of gamma diversity and mean alpha diversity, and varies between 1 (when species are the same in all sampling units), and the number of sampling units (when all species are different). Because different scales with different numbers of sampling units were compared, CS qCS as a direct transformation of beta diversity qβ was used: qCS=(1/qβ−1/N)/(1−1/N)

This converts beta diversity into values between 0 (no similarity) and 1 (complete similarity). More generally speaking, qCS-values below 0.33 were considered as low and values above 0.66 as high similarity. qCS for q = 0 and N = 2 equals the Jaccard index, and for q = 2 the Morisita–Horn index.

Entropart package in the R-program for diversity partition was used (Marcon & Herault, 2015; R-Development-Core-Team, 2009). Chao 1 richness estimator, as well as sampling coverage, was calculated in order to address sampling completeness for each mountain (Colwell, 2010; Chao & Jost, 2012).

Homogeneity of multivariate dispersion using Jaccard and Morisita–Horn dissimilarity indices was carried out (Anderson, 2001) applying permutation test based on 999 repetitions and Tukey´s honest significant difference method with the vegan package in R (Oksanen et al., 2007; R-Development-Core-Team, 2009). Sampling units were the 16 sites, grouped by mountains or by three FSP mosaic classes (see Supplemental Information).

Species were classified regarding their DP as: NT, Mexican High Plateau (NT origin), Meso-American Montane (MM, species evolved in Mesoamerican mountains but of both northern and southern origin), Nearctic (of recent northern origin), and Paleoamerican (which Halffter divided into the subpatterns Mountain Paleoamerican, Paleoamerican High Plateau and Tropical Paleo-American; Halffter, 1964, 1976, 1978).

Results

About 3,430 individuals of 29 species at the four mountains were collected during the three sampling seasons (see Table 1). The most diverse subfamily was Aphodiinae with 16 species in 10 genera, followed by Scarabaeinae (eight species in three genera) and Geotrupinae (five species in three genera). The most abundant subfamily was Scarabaeinae with 60% of all individuals, followed by Aphodiinae and Geotrupinae (25% and 15%, respectively). These percentages are nearly the same across altitudes and sites. Regarding different altitudes, 25 species were collected at lower altitudes (with ten unique species) versus 19 species at superior altitudes (with four unique species). Of the 29 species, five were captured at all volcanoes; seven species were restricted to two and 13 were unique to one of the four volcanoes. Sampling completeness was high, values of Chao 1 estimators were 92% for the pooled mountains, and 96%, 81%, 89 %, 78% for each of the mountains (MA, CP, PO, and SN respectively), and sampling coverage varied between 98.7% and 99.9% (see Table 1).

Table 1 Species sampled at each volcano.

Species	SF	DP	MA	CP	PO	SN	Four volcanos	
Agrilinellus azteca	A	PM	0.2	8.8	8.3	5.4	5.71	
Agrilinellus ornatus	A	PM*	2.7	0.5	4.0	17.9	6.27	
Blackburneus charmionus	A	NT*			0.1		0.02	
Blackburneus guatemalensis	A	NT*	0.1		2.8		0.73	
Blackburneus saylorea	A	NT*			0.2		0.04	
Cephalocyclus hogei	A	NA		24.0	3.1	0.4	6.86	
Gonaphodiellus bimaculosus	A	NT			1.3		0.33	
Gonaphodiellus ophisthius	A	NT	1.6	0.2	8.4		2.54	
Labarrus pseudolividus	A	NT				0.4	0.10	
Neotrichonotulus inurbanus	A	NA		0.1			0.02	
Oscarinus indutilis	A	NA		0.1	0.1		0.04	
Oxyomus setosopunctatus	A	NA			1.3		0.33	
Planolinellus vittatus	A	NA		0.1	0.4		0.12	
Trichonotuloides alfonsinae	A	PM*			0.5	0.8	0.33	
Trichonotuloides hansferyi	A	PM*		0.1			0.02	
Trichonotuloides glyptus	A	PM		1.0			0.26	
Ceratotrupes bolivari	G	PM	3.6	0.8	0.1	0.4	1.23	
Halffterius rufoclavatus	G	PM		2.1	0.3		0.60	
Onthotrupes herbeus	G	PM	2.3	1.0	24.3	0.8	7.09	
Onthotrupes nebularum	G	PM	0.6	0.2	8.8		2.38	
Onthotrupes sallei	G	PM*		0.4			0.11	
Copris armatus	S	PM	11.9			1.7	3.39	
Onthophagus aureofuscus	S	PM		9.9	1.1		2.74	
Onthophagus bolivari	S	PM*	20.9				5.22	
Onthophagus ch. chevrolati	S	PM	38.8	50.2	34.8	72.1	49.00	
Onthophagus lecontei	S	HP	15.4	0.6			4.00	
Onthophagus mexicanus	S	HP	1.3				0.33	
Phanaeus qu. quadridens	S	HP	0.5				0.12	
Phanaeus a. amethystinus	S	MM			0.2		0.04	
Individuals			824	1,155	1,211	240	3,430	
Richness			13	17	19	9	29	
Chao 1/Coverage (%)			96/99.9	81/99.7	89/99.7	78/98.7	92/99.9	
Notes:

Species sampled via pit fall traps, indicating relative abundance percentage for species at each volcano. Values for four volcanos are mean %. SF, subfamily; DP, dispersion pattern (HP, high plateau; MM, mesoamerican montane; NA, nearctic; PM, Paleoamerican; NT, neotropical).

* DP unknown but assumed according to the genera. Volcanos: MA, Malinche; CP, Cofre de Perote; PO, Pico de Orizaba; SN, Sierra Negra.

Variability was the most consistent pattern at all scales and measures (dung beetle diversity, abundance, and CS, see Figs. 2 and 3). Among and within all mountains, there were a high variation of diversity and abundance values (Fig. 2): Richness varied between 2 and 15 species and from 1.1 to 4.7 effective species for order q = 2. The abundance ranged from 9 (SN upper leeward site) to 585 individuals (PO lower western site). On the same mountain, measures could vary between sites by three-fold (0D at PO, 2D at CP) to as much as 24-fold (abundance at MA). At each mountain, the forest-dominated sites did not have higher or lower diversity, abundance or CS values when compared to the sites with a FSP mosaic, and were similarly variable (Figs. 2 and 3; Fig. S1).

Figure 2 Alpha and gamma diversity of the four mountains.

Results of alpha and gamma diversity for orders q = 0 and q = 2 (upper and lower value, respectively), and abundance for each sampling site. Values in the central square show gamma diversity of each mountain. Numbers below the square indicate abundances. (A) MA, La Malinche; (B) CP, Cofre de Perote; (C) PO, Pico de Orizaba; (D) SN, Sierra Negra. W, western leeward side; E, eastern windward side. Upper squares display values at 3,400 MASL and lower squares at 2,700 MASL.

Figure 3 Compositional similarity at the four mountains.

Compositional similarity qCS within and among sites (q = 0 upper value, q = 2 lower value) at the four mountains: (A) MA, La Malinche; (B) CP, Cofre de Perote; (C) PO, Pico de Orizaba; (D) SN, Sierra Negra. Values inside the squares show CS among traps. Left squares—western leeward sides, right squares—eastern windward sides. Upper squares display values at 3,400 MASL and lower squares at 2,700 MASL. Values without squares show the pairwise CS between sites, arrows indicate which pairs were compared.

Comparing mountains, PO had the highest diversity and abundance. CP presented the second highest species richness and abundance, and the lowest CS. SN had by far the lowest alpha and gamma diversity (less than 50% of adjacent PO), as well as abundances (20% compared to PO, see Fig. 2), whereas CS varied similarly as in MA and PO. At CP, the relatively high gamma diversity was product of the high beta diversity. CS demonstrated differences among and within mountains depending on the order q. CP had the lowest CS for both q = 0 and 2. Sites at Sierra Negra shared only a few species, but the abundant species were most similar, so 0CS was nearly as low as at CP, but 2CS-value was the highest of all mountains. MA had the highest CS values for order q = 0 (although they were not high), and second highest for order = 2.

Compositional similarity between pairs of mountains did not increase with distance (Fig. 4). Species similarity was relatively low among mountains (0.3–0.38), with the exception of the 0CS between CP and PO (0.5).

Figure 4 Compositional similarity between pairs of mountains in relation to its distances.

Compositional similarity and distances between each pair of mountains, the upper value presents 0CS, the lower value 2CS (Jaccard and Morisita–Horn compositional similarity, respectively). The table indicates CS-values when the most abundant species Onthophagus chevrolati chevrolati (with a mean relative abundance of 0.5), was excluded from the analysis (see text). MA, Malinche; CP, Cofre de Perote; PO, Pico de Orizaba; SN, Sierra Negra. Positions and distances based on Google earth Images.

The Scarabaeinae Onthophagus chevrolati chevrolati Harold dominated all mountains, with 50% of all collected individuals, and its distribution pattern was principally responsible for the higher 2CS-values between all mountains (see Fig. 4). 2CS was the highest between CP and PO with 0.84, meaning that similarity was greater between these two mountains at a 53 km distance than between PO and SN at a distance of 7 km. The great dominance of O. ch. chevrolati hid the high variability of the other species, as shown in the table in Fig. 4. When omitting O. ch. chevrolati from the analysis, 2CS turned into low values, and in general, it decreased as distance increased.

The permutation test for beta diversity did not reveal significant differences in variability among mountains, aiming that, as already described as the general diversity pattern for our study, variability was similarly high at all mountains (Fig. S1).

Temporal variation was also high; each year approximately 60% of overall richness was collected (Table S1). The most abundant species, O. ch. chevrolati was present at each mountain every year. In contrast, eight species were present only at one mountain one year. Most species also varied highly regarding abundance between years, inclusive at the same mountain. Each year, every mountain displayed different richness and CS (see Tables S1 and S2).

Dispersion patterns exhibited differences among mountains; for instance, species representing the Nearctic Pattern were absent in MA but abundant in CP and present in PO. Pico de Orizaba was the only mountain with one Mesoamerican Montane Pattern species. Paleoamerican Mountain pattern dominated both at species and individual level, at all mountains. SN was almost entirely dominated by individuals of PM-species (Fig. S2).

Regarding taxonomic differences, MA had by far the highest percentage of Scarabaeinae individuals and species. On the other three volcanoes, Aphodiinae presented the highest species richness, while Scarabaeinae had the highest abundance. Geotrupinae showed the highest variation in abundance, being most abundant on PO and least abundant on the adjacent SN (Fig. S2).

Discussion

As predicted, this study provides evidence that high mountains can hold a diverse dung beetle community. Unexpected and contrary to our predictions, we found high variability in diversity patterns at different spatial scales, both within and between mountains. No previous studies on dung beetles in Mexico have reported species richness numbers for these elevations: Martín-Piera & Lobo (1993) collected five species above 2,500 MASL; Lobo & Halffter (2000) collected seven species; Escobar, Halffter & Arellano (2007) only two; Halffter et al. (2008) eight species; and Alvarado, Escobar & Montero-Muñoz (2014) four species (Halffter et al. (2008) did not report Aphodiinae and Escobar, Halffter & Arellano (2007) only Scarabaeinae). Of these, only Lobo & Halffter (2000) collected three species at 3,300 MASL, whereas 19 species were found in this study.

More than 10% of species collected were new to science (Arriaga-Jiménez et al., 2016; Dellacasa, Dellacasa & Gordon, 2014). The three new species belong to the Paleoamerican mountain DP, which Halffter & Morrone (2017) described as corresponding to lineages that have undergone vicariant speciation. The discovery of Onthophagus bolivari, which probably evolved in the mountain complex of La Malinche and El Pinal (Arriaga-Jiménez et al., 2016) highlights the importance of the O. chevrolati species group, whose distribution reflects the MTZ and which now comprises more than 40 species and subspecies (Zunino & Halffter, 1988). In addition, some species at high mountains developed particular feeding behaviors, living exclusively in gopher nests (Rodentia: Cratogeomys), feeding on excrements stored in latrines (Zunino & Halffter, 1988, 2007). Four dung beetle species (Onthophagus hippopotamus, Geomyphilus pierai, G. barrerai, and Neotrichonotulus perotensis, see Table S2) were collected only in the gopher nests, as they were not attracted to bait (thus explaining why they could not be included in the diversity analysis), but they also support the conclusion that high mountains are a center of speciation associated with a unique, highly adapted dung fauna.

Dispersion processes may be the main driver of dung beetle richness and composition in mountains, as only some species are restricted to one or few mountain chains. Lobo & Halffter (2000) discussed how mountains of the TMVB illustrate horizontal colonization—an observation that is indeed supported by the DP of most species collected in this study. As observed by Caballero et al. (2011), temperature and vegetation equivalents in the last glacial maximum were distributed 1,000 m below the current altitudes. As a consequence, these conditions were not limited to mountain tops as is the case today, but instead connected over large areas. As there have been more glaciation periods in the MTZ, these habitat replacements with repeated separations and a mix of faunas might have been common (Zunino & Halffter, 1988). Halffter & Morrone (2017) stated that in the MTZ vertical colonization occurred rarely in relation to horizontal colonization and in comparison to other tropical regions. Nevertheless, several species with NT DP (Lobo & Halffter, 2000 only reported one species) show that vertical colonization events also contributed to species richness and composition in the TMBV (Lobo & Halffter, 2000).

Whereas speciation and DP may explain dung beetle richness and composition of the four volcanoes studied, they do not explain the high variability of diversity and CS patterns found at different spatial scales.

Volcanoes in the TMVB can be considered as dynamic, ecologically instable systems with a comparably strong disturbance regime at different time scales. Volcanism, earthquakes, and glaciations cause long-term temporal effects of habitat destruction and perturbation, leading to local extinctions and population splitting, followed by speciation, and recolonization (Zunino & Halffter, 1988; Siebe et al., 1996; Castro-Govea & Siebe, 2007; Battisti, Poeta & Fanelli, 2016).

Severe climate, diverse soil conditions, and the resulting FSP mosaic form heterogeneous conditions of permanent environmental instability. Because dung beetles are mainly linked to herbivorous mammals, they may be especially vulnerable to environmental instability (Nichols et al., 2009). It has often been reported that diversity patterns produced by instability are high regional diversity, due to lower local (alpha) diversity but higher beta diversity (Halffter et al., 2007; Martínez et al., 2015; Battisti, Poeta & Fanelli, 2016). While our study provides some support for such patterns (e.g., for Cofre de Perote), instability seems nevertheless to lead to highly variable patterns: high and low abundance, diversity and CS, without any apparent linkage to measured ecological variables. Also, the dominance of O. ch. chevrolati may hide a clearer pattern of the expected low similarity; it is not common that one species dominates a dung beetle community to such a degree that similarity patterns change totally when it is excluded from analyses. O. ch. chevrolati, also present in other studies of the region (Lobo & Halffter, 2000; Escobar, Halffter & Arellano, 2007), is abundant in all mountains of the TMVB in forest and pastures, and it is the species with the highest known elevation record in Mexico at 3,800 MASL (Zunino & Halffter, 1988). Nevertheless, there is little information about its natural history or physiological adaptations to high mountain climates and habitats. This species belongs to a recent lineage in an ongoing phase of dispersion (Zunino & Halffter, 1988). As shown in the results, when excluding O. ch. chevrolati from CS analysis for order q = 2 (regarding abundant species), the community presented a large degree of heterogeneity and dynamism. Even the remaining most frequent species had dissimilar abundances on each mountain, indicating that dynamics are independent and different on each volcano. Severe climate and environment could cause high population fluctuations, where low densities of individuals result in a lower species detectability (MacKenzie et al., 2003). Alternatively, we found some species to be active only in a small window of time due to their biological cycles (e.g., Cephalocyclus). Our temporal data indicate high fluctuations from year to year and between volcanoes for these species, although they could not be linked directly to climate data.

The observed patterns of diversity and heterogeneity could be typical for the entire MTZ, with lower values of diversity and heterogeneity near its northern and southern limits, and more endemisms in the southern part of the zone (Zunino & Halffter, 1988). As elevation increases, the species turnover is predicted to be higher: Scarabaeinae richness (a temperate and tropical group), will be replaced by species of Aphodiinae (a heterogeneous group with different origins) and Geotrupinae (a northern group of colder climates). Because of its origins, Geotrupinae will lose importance southwards, whereas Aphodiinae and Scarabaeinae will present still high richness and abundance. By contrast, in the NT region, Geotrupinae cannot compete with Scarabaeinae (Hanski & Cambefort, 1991), and are absent in high mountains. Scarabaeinae richness decreases at higher mountain altitudes, sharing only some species with the MTZ (Escobar et al., 2007; Halffter & Morrone, 2017; Howden, 1964). Distribution of Aphodiinae species are not well documented for high mountains, so it is not clear if they show similar richness patterns as in the TMBV, or if they are mostly absent. The fact that dung beetles are frequently studied in altitudinal gradients may erroneously lead to the assumption that their high mountain diversity patterns are well understood. This work reveals the need to conduct more extensive studies at high elevations in tropical montane systems to understand some of these as-yet unresolved issues surrounding dung beetle diversity, biogeography, and evolution.

Present conservation strategies do not protect all the high mountain fauna since reserves are situated mostly at higher altitudes. New proposals such as archipelago reserves (Halffter, 2005, 2007) may be an adequate tool, given mountain tops have limited areas with similar ecological conditions and similar threats (land use intensification, climate change). These systems would also benefit from coordinated monitoring and conservation programs.

Future studies should prioritize representative sampling of other mountains in the MTZ. Volcanic mountains, dominant from Mexico to Central America, should be compared to mountains of different origin. For instance, some regions in the adjacent Mexican states of Oaxaca or Chiapas had different disturbance patterns over a large temporal scale, which could lead to different diversity patterns. South America also has a heterogeneous mix of high mountains with and without volcanic activity, so there may be differences between them and Mexico. More importantly, though, the biogeographical history and DP in the Andes differ strongly from those of the MTZ. Less is known about Asian tropical high mountains, despite the fact that all three dung beetle groups are present there. Filling in some of these research gaps will allow comparison of these mountain regions and will improve our understanding of species diversity and turnover in high mountain ecosystems. Despite the research needs that remain, our study nevertheless makes progress by providing evidence of how tropical high mountains can improve our understanding of the drivers of diversity patterns, and how biogeographic history and ecological factors mold them. This knowledge is important to improve the conservation of these unique and restricted ecosystems.

Supplemental Information

Supplemental Information 1 Supplementary material (2 Figures, 2 Tables).

Click here for additional data file.

Supplemental Information 2 Number of individuals per trap.

Click here for additional data file.

We thank Fernando Escobar (Instituto de Ecología, A.C.) for his valuable aid for field work and curatorial work for the beetles collected. Three anonymous reviewers provided valuable comments which helped us to improve the manuscript. We are grateful to Maria del Sagrario Velasco García and Mauricio Rodriguez Herrera, who revised the English version of this manuscript.

Additional Information and Declarations

Competing Interests

Author Contributions

Field Study Permissions

Data Availability

The authors declare that they have no competing interests.

Alfonsina Arriaga-Jiménez conceived and designed the experiments, performed the experiments, analyzed the data, contributed reagents/materials/analysis tools, prepared figures and/or tables, authored or reviewed drafts of the paper, approved the final draft.

Matthias Rös analyzed the data, contributed reagents/materials/analysis tools, prepared figures and/or tables, authored or reviewed drafts of the paper, approved the final draft.

Gonzalo Halffter conceived and designed the experiments, contributed reagents/materials/analysis tools, authored or reviewed drafts of the paper, approved the final draft.

The following information was supplied relating to field study approvals (i.e., approving body and any reference numbers):

Field experiments were performed with a permission of the Secretaria del Medio Ambiente y Recursos Naturales, Mexico (FAUD-0018).

The following information was supplied regarding data availability:

The raw data is provided as Supplemental Dataset Files.

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
