# Peer review of "High variability of dung beetle diversity patterns at four mountains of the Trans-Mexican Volcanic Belt"

_PeerJ, doi:10.7717/peerj.4468_

## Round 0.1 · original submission · Major Revisions

All three reviewers have provided substantive comments that need to be addressed. Before re-submitting pleas make sure the language used is much clearer. Please find a strong native English speaker to carry out a thorough edit of the manuscript.

Reviewer 1 ·

Basic reporting

Some language polishing could improve the manuscript. Some of the lines where English could be improved are presented with structure (manuscript line: language item -> improvement suggestion):
Line 100: template -> temperate
Line 139: “more” than which other forest?
In general a more scientific style could improve the quality of the manuscript.

The manuscript is generally sufficiently referenced. However some references could be added to improve the contextualization (mainly in the introduction):
Line 117: The authors claim that their work is the first study on diversity patterns at high altitudes. However, there is a paper on dung beetles from 0 to 4000m. See Herzog et al. 2013. However, Herzog et al. study database material, it is not a field study. Moreover it focuses in Phanaenini, not including Aphodiinae neither Geotrupidae.
Surprisingly in the first paragraph of the discussion (line 255) they cite several references that report diversity patterns at similar altitudes.


Data is shared as supplementary material.

Reference:
Herzog S.K., Hamel-Leigue A.C., Larsen T.H., Mann D.J., Soria-Auza R.W., Gill B.D., Edmonds W., & Spector S. (2013) Elevational distribution and conservation biogeography of Phanaeine dung beetles (Coleoptera: Scarabaeinae) in Bolivia. PLoS One, 8, n

Experimental design

The manuscript lacks specific aims. Accordingly, hypothesis are missing and could be included to have a more aim-oriented manuscript. This results in doubts about the interest of the study.

Such a lack of focus and/or apparent interest stands out in the experimental design, which is my main concern of this work. Sampling sites seem loosely placed, which leads to a high variability of habitats in which they sample, forcing analysis to be over complicated for the main aim of the study.

However, methods used (beta, PCoA and TukeyHSD) seem appropriately chosen, explained and implemented.

Validity of the findings

Data is controlled, well replicated and robust, although a high proportion of traps did not yield any capture, which is understandable at such altitudes. Moreover, sampling completeness and coverage was adequately estimated (however, using the new method described in Hsieh et al. 2016 is recommended in the future).

Results are discussed in a generally speculative manner, not very linked to the reported results. I would encourage the authors to explain the patterns observed in a more contemporary scenario, rather than a broad geographical scale (South America mentioned several times) and a geological time scale (thousands and millions of years argued).

Reviewer 2 ·

Basic reporting

This manuscript offer the results of dung beetle surveys carried out in four Mexican mountains of the Trans Mexican Volcanic Belts (TMVB). These results are interesting because the available information about these assemblages is scarce and these surveys provide faunistic information in mountains not previously surveyed. However, the authors need to offer more exhaustive information about former published studies carried out in the TMVB with dung beetles, clearly discriminating the novelty of their results within the context of the results and explanations provided by these former studies. Please consider that some of the authors of this manuscript also participate as co-authors in these former studies. Similarly, I consider that the manuscript need to offer clear objectives rooted in the available knowledge about temporal segregation or the meaning of the difference in the “dispersal patterns” of species. The authors state that “We aimed to describe diversity patterns within and between mountains and to explain the origin of these patterns, focusing on biogeographical (e.g. vertical and horizontal colonization, and speciation), as well as ecological variables (e.g. elevation, exposure, forest mosaics)”. In the discussion authors interpret the obtained results so that the main purpose of the study is not only descriptive. Authors should indicate what the implications are if richness and compositional patterns between mountain chains are similar or different, and also what is the purpose/implications of a variation (or lack of it) in the dispersal pattern.
• Abstract. “our results indicate that species richness at high mountains”. Probably it is better to indicate that “our results indicate that both species richness and composition at high mountains”
• The last sentence in the abstract is in my opinion not directly linked with the results of the manuscript and could be removed.
• Line 52. The concepts of vertical and horizontal colonization are not standard in the literature and should be described in the introduction.
• Line 77. “The mountains in the MTZ neither were glacial refuge…”. This statement is relatively contradictory with the own results provided by the authors and the existence of endemic species in these mountains belonging to Holarctic genera.
• Line 99. “Whereas Scarabaeinae are most diverse and abundant in the tropics, the two latter groups are more diverse in northern template regions”. This statement should be supported by a reference.
• In the introduction there are some unnecessary new paragraphs (line 81, line 112) which explain questions related with former paragraphs.
• Lines 112-115. Please add some references supporting these affirmations
• Line 116. “we present the first study of diversity patterns of dung beetles carried out at altitudes between 2700 and 3400 MASL”. There are other elevational studies using dung beetles in the Neotropical mountains at similar altitudes, some of them in Mexico, and the authors need to acknowledge and reference these studies.
• Line 121. Change “origin” by “probable origin”
• Consider to change directions by slopes
• Fig. 3. According to the legend, values inside squares represent similarity among traps. However, if only a trap is located in each sampling occasion this in fact represent temporal or seasonal variation.
• Line 229. All these compositional comparisons lack statistical support and the permutational test shows that there are not differences. The same happens in the case of “dispersion patterns” (dispersion or dispersal?; see Systematic Biology 26: 210-211)
• Line 260. In the paper of Lobo & Halffter (2000) there are 13 species cited above 300 MSl, not three.
• Line 254. Cofre de Perote can be considered not well sampled?

Experimental design

• Line 120. If the study of temporal dynamics constitutes an objective of this study, authors need to incorporate in the introduction the status of the available knowledge about this question.
• Methods. I suggest including topography in the map locating each one of the four sampled mountains.
• Please define clearly what is the used sampling unit in subsequent analyses (e.g. all the specimens collected in the pitfall traps of an altitude and slope independently of the month)
• Line 164. Please specify the sampling months.
• Line 170. Please explain why direct observations about the dung beetles inhabiting gopher nest are important, and also why these results are not included in the analysis.
• Methods. It is necessary to provide additional information about the followed nomenclature as well as how these specimens have been taxonomically determined. The identification of Aphodiinae species is not easy and the lack of comparative material seriously difficult this task. Please also indicate where voucher specimens are located.
• Line 183. Please specify how beta diversity is calculated. If you have two altitudes and two slopes, is beta diversity reflecting the variability promoted by these two factors (slope and altitude)?
• Analysis. The iNEXT online package (http://chao.stat.nthu.edu.tw/wordpress/software_download/inext-online/) can be used to estimate the probable number of species of each assemblage and the degree of completeness of collected inventories. In my opinion, authors need to demonstrate that the compared inventories are all submitted to similar sampling efforts, and iNEXT allow us to calculate the “coverage” reached for each assemblage because comparisons need to be carried out when estimated diversities have a common sample completeness” (see Ecological Monographs, 84, 45–67).
• Table 1 must show “real” absolute number and not relative abundance values. Furthermore, authors should show the number of individuals collected for each species and site according with slope of each mountain.
• Line 208. This is the first indication about sampling completeness. This question should be explained in the methods and the use of iNEXT may guarantee that comparisons are reliable. If completeness values oscillate from 78% to 96% the species richness comparisons may be erroneous because undetected species are not considered.
• Line 210. These analyses need to be described in methods.
• From Line 215. Such comparisons should consider undetected species to be valid (see Chao et al. 2017. Ecology in press, and Good Turing test at http://chao.stat.nthu.edu.tw/wordpress/software_download/good_turing/)
• Line 241. If the authors aim to examine temporal variation in each site will be necessary to demonstrate that the inventories obtained in each period are comparable and exhaustive. Please consider that you have only one pitfall trap by month, altitude and slope.

Validity of the findings

no comment

Additional comments

no more comments

Reviewer 3 ·

Basic reporting

Improving the English used throughout the paper would greatly improve clarity. In particular the syntax of certain sentences make comprehension difficult:
• Lines 24-26: Reword
• Lines 31-32: Reword
• Line 54: “refugees” should be “refuges”
• Line 71: Reword this sentence
• Lines 74-77: Reword this sentence to be more concise and make the overall point clearer
• Line 82: missing degree symbol from “19”
• Line 83-84: I’m unclear on the meaning of this sentence. Does “peak elevation” refer to the height of the eight highest volcanoes, the height of all mountains in the region, or the highest point of the highest point of the high plateau? To me “peak elevation” suggests the highest point of the plateau, but the fact that a range is given indicates on of the other two options.
• Lines 94-96: Reword
• Line 100: should be “Temperate”
• Lines 100-115: This whole section could be reworded to make the overall point clearer
• Lines 116-120: Reword this as its unclear just what makes this study unique. Is it the first study to look at dung beetle diversity solely in the TMVB? Is it the first to look at diversity between 2700 and 3400 MASL? Is it the first to look at diversity at these four mountains? Or is it some combination of these?
• Line 135: should these elevations be 2700 and 3400 MASL?
• Line 141: “…patches where trees are at an early….”
• Lines 142-148: Does this paragraph refer to the sites in the TMVB?
• Line 204: “…nearly the same across altitudes…”
• Line 209: “…mountains…”
• Lines 212: This sentence can be removed as the content is covered in the subsequent paragraphs.
• Line 242: “…O. ch. Chevrolati…”
• Line 264: “…These mountain communities…”
• Line 294: “…O. ch. Chevrolati…”
• Line 296: “…O. ch. Chevrolati…”
• Line 201: “…O. ch. Chevrolati…”
• Line 316-317: Reword this sentence

Minor notes
• Lines 54-57: Provide references for this statement
• Lines 100-102: move references to the end of the sentence
• Lines 112-115: Provide references
• Line 129-134: Including the elevations of all of the mountains would be good
• Lines 142-148: Provide references
• Line 165: reference Larsen & Forsythe, 2005 for trap spacing?

Experimental design

The methodology of the study is generally well described however the addition of some minor details would be appreciated.
• Line 164: What months of the year does the rainy season occur?
• Line 165: What type of dung was used?
• Line 165: I’m a little unclear on how the trapping was repeated. Was trapping undertaken twice at each site, per year, e.g. in June 2001 traps at each of the four sites on MA were baited and left for 48 hours twice?
o How far apart were these two sampling dates?

Validity of the findings

The content of the discussion is predominately speculation. While this speculation is relevant to the questions asked and well referenced there should be greater emphasis on addressing the stated aims of the study.
• Lines 276-277: While this statement may be true without some more context regarding the number of pholephile species found outside of mountain areas it doesn’t seem to be borne directly from the results.
• More discussion concerning the effect, or lack-thereof, of the ecological variables
• More discussion on the biogeographic affiliation of the species and patterns of dispersion, given that these are emphasized throughout the paper
• Some discussion on the temporal variation found would be welcome especially considering the large differences in the abundances of some species in 2011 compared to subsequent years seen in table S1

Minor notes
• Lines 203-204: these percentages add up to 101%
• Lines 206-207: can the specific volcanoes be given
• Line 207: the fact that 13 species were found at only one of the four volcanoes is repeated later and can be removed.
• Line 237: The abstract states that once O. ch. Chevrolati was removed from the analysis there was a positive relationship between distance and CS however in the results it states that CS decreased with distance.

Additional comments

The paper presents a number of new and interesting findings, is well referenced and the figures are clear and informative, however it suffers due to poor clarity and a lack of focus. In particular greater emphasis on describing and discussing the aims as stated in the paper would be good.

---

## Round 0.2 · Minor Revisions

The reviewers are all happy with your revisions. Some minor issues still need to be resolved.

Reviewer 1 ·

Basic reporting

The clarity of the text has greatly improved from he previous version. I congratulate the authors for the effort.
Several references have been added to support some statements and the overall importance of the study as filling a knowledge gap.
Figures, tables and raw data comply with the journal standards.
Results are relevant for the aims and predictions.

Experimental design

Aims of the study meet the scope of the Journal.
This second version defines the aims of the study much better than the first version, fulfilling one of the main weaknesses pointed out by the three reviewers.
Methods are explained with enough detail for the study to be reproducible.

Validity of the findings

Data is statistically sound and perfectly controlled with several tools, perhaps even to much after reviewers demand.

Speculation in the discussion (a point in which the three reviewers commented) is much better supported with literature and results in the study.

Additional comments

NOT to be taken as a fault in this study but as a scientific thought:
I continue to wonder if other sampling sites could have been chosen (or could be studied in the future if more funds are available). This idea comes to my mind specially thanks to the greatly improved figure/map including elevation of the whole Volcanic Belt. The map clearly shows that there are at least 7 isolated mountain ranges in the Volcanic Belt, of which only 2 are sampled in this study. Moreover, these are the most eastern ones, leaving 5 mountain ranges not sampled.
I wonder if the patterns found in this study can be generalized to the whole Volcanic Belt or even if a more homogeneous pattern would be found (dropping the idea of the "high variability" mentioned throughout the manuscript and even the title).
I reinforce the idea that my suggestion does not lessen the value of this study, it just expresses my eagerness to see this research continue with further sampling in the other mountain ranges in Mexico (and elsewhere).

Reviewer 2 ·

Basic reporting

I acknowledge the efforts of the authors to include in the new version most part of my former suggestions and criticisms. In the current version I only have some minor proposals now included as notes in the PDF version of the manuscript. If the authors agree to them, I suggest the acceptance of the manuscript without the need for a subsequent revision.

Experimental design

I now only suggest alternative definitions of vertical and horizontal colonizations

Validity of the findings

OK

Additional comments

I acknowledge the efforts of the authors to include in the new version most part of my former suggestions and criticisms. In the current version I only have some minor proposals now included as notes in the PDF version of the manuscript. If the authors agree to them, I suggest the acceptance of the manuscript without the need for a subsequent revision.

Annotated reviews are not available for download in order to protect the identity of reviewers who chose to remain anonymous.

Reviewer 3 ·

Basic reporting

Some minor typos:
• Line 88: reference
• Line 277: missing “of”?

Experimental design

no comment

Validity of the findings

no comment

Additional comments

The readability and flow of the paper is greatly improved. The introduction is well referenced and provides a good overview of the concepts underpinning the research as well as the important geographical context. Holes in the literature are identified and the importance of filling these holes well justified. The results of the study lend support to many of the hypotheses put forward and demonstrate the importance of sampling effort when surveying montane communities. The high degree of variability between communities combined with the dominance of O. ch. chevrolati was extremely interesting, particularly as it was found in both forests and pastures.

---

## Round 0.3 · accepted · Accept

I am happy with the changes you have made - I do hope the manuscript is well received!